



# Deployment of the C-band radar Poldirad on Barbados during EUREC[4]A

Martin Hagen[1], Florian Ewald[1], Silke Groß[1], Lothar Oswald[1], David A. Farrell[2], Marvin Forde[2], Manuel Gutleben[1,6], Johann Heumos[3], Jens Reimann[4], Eleni Tetoni[1], Eleni Marinou[1,5], Gregor Möller[6], Christoph Kiemle[1], Qiang Li[1], Rebecca Chewitt-Lucas[2], Alton Daley[2], Delando Grant[2], and Kashawn Hall[2]

[1]Deutsches Zentrum für Luft- und Raumfahrt, Institut für Physik der Atmosphäre, Oberpfaffenhofen, Germany
[2]Caribbean Institute for Meteorology and Hydrology, Husbands, St. James, Barbados
[3]Deutsches Zentrum für Luft- und Raumfahrt, Objektbewirtschaftung, Oberpfaffenhofen, Germany
[4]Deutsches Zentrum für Luft- und Raumfahrt, Institut für Hochfrequenztechnik und Radarsysteme, Oberpfaffenhofen, Germany
[5]National Observatory of Athens, Institute for Astronomy, Astrophysics, Space Applications and Remote Sensing, Athens, Greece
[6]Ludwig-Maximilians Universität, Meteorologisches Institut, München, Germany

**Correspondence:** Martin Hagen (martin.hagen@dlr.de)

**Abstract.** The German polarimetric C-band weather radar Poldirad (Polarization Diversity Radar) was deployed for the international field campaign EUREC[4]A (ElUcidating the RolE of Cloud-Circulation Coupling in ClimAte) on the island of Barbados. Poldirad was operated on Barbados from February until August 2020. Focus of the installation was monitoring clouds and precipitation in the trade wind region east of Barbados. Different scanning modes were used with a temporal sequence of 5 minutes and a maximum range of 375 km. In addition to built-in quality control performed by the radar signal processor, it was found that the copoloar correlation coefficient $\rho_{HV}$ can be used to remove contamination of radar products by sea clutter. Radar images were available in real-time for all campaign participants and onboard of research aircraft.Examples of mesoscale precipitation patterns, rain rate accumulation, diurnal cycle, and vertical distribution are given to show the potential of the radar measurements for further studies on the life cycle of precipitating shallow cumulus clouds and other related aspects. Poldirad data from the EUREC[4]A campaign are available on the EUREC[4]A AERIS database: https://doi.org/10.25326/218 (Hagen et al., 2021a) for raw data and https://doi.org/10.25326/217 (Hagen et al., 2021b) for gridded data.

## 1 Introduction

Clouds and precipitation play an important role in the Earth radiation budget. Predictions for future climate scenarios show large uncertainties in the contribution of clouds on the overall radiation budget. One considerable uncertainty can be attributed to shallow maritime cumulus clouds as they are frequently observed over tropical oceans. Long-term continuous observations





are rare and only a limited number of detailed measurements are available to foster understanding of their development, life cycle, and organization. Satellite observations can provide the structure of cloud patterns – mainly through high-resolution visible images during daytime – but rarely can describe the thermodynamic and dynamic environment in which the cloud

pattern does exist. Because of the lack of continuous observations, various field campaigns have been performed in the Atlantic trade wind region to investigate in detail the role of shallow maritime cumuli and their relation to air-sea interaction and global circulation: BOMEX 1969 (e.g. Fleagle, 1972; Holland and Rasmusson, 1973); GATE 1974 (e.g. Kuettner et al., 1974; Kuettner and Parker, 1976); ASTEX 1992 (Albrecht et al., 1995); RICO 2004/05 (e.g. Rauber et al., 2007). In order to further elucidate the thermodynamic and dynamic environment of maritime clouds and their organization the EUREC[4]A (ElUcidating

the RolE of Cloud-Circulation Coupling in ClimAte) campaign was planned (Bony et al., 2017) and performed in January and February 2020 in the trade wind zone east of Barbados (Stevens et al., 2021).

One property of maritime cumulus clouds in the tropics is a rapid initiation of rain in the shallow cloud systems with cloud top heights in the range of 2.5 to 5 km (e.g. Rauber et al., 2007). The development of a cumulus cloud to a raining convective cloud occurs within a time span of about half an hour (e.g. Saunders, 1965). Detailed radar studies were conducted during the

RICO campaign (e.g. Nuijens et al., 2009; Snodgrass et al., 2009; Trivej and Stevens, 2010), and showed the importance of long range surveillance radar observations with high spatial and temporal resolution. In this view the full polarimetric C-band research radar system Poldirad (Schroth et al., 1988) was deployed for EUREC[4]A on the island of Barbados. A non-polarimetric S-band radar of Barbados Meteorological Service was out of service during the EUREC[4]A campaign. Thus Poldirad was the only radar system which could provide an overview over precipitation fields during the campaign.

The EUREC[4]A campaign, its scope, and the role of Poldirad is introduced in Sect. 2, Poldirad and data processing is described in Sect. 3. Section 4 gives some measurement examples. Section 5 describes the data available on the EUREC[4]A AERIS database, and Sect. 6 concludes the study.

## 2   EUREC[4]A campaign

EUREC[4]A is an international initiative in the scope of the World Climate Research Programme's Grand Science Challenge

on Clouds, Circulation and Climate Sensitivity. The field phase of EUREC[4]A took place between 20 January and 20 February 2020 in the region east of Barbados (Bony et al., 2017; Stevens et al., 2021).

EUREC[4]A aims to advance the understanding of the interplay between clouds, convection and circulation and their role in climate change. Scientific questions are (see Bony et al., 2017): (i) How resilient or sensitive is the shallow cumulus cloud amount to variations in the strength of convective mixing, surface turbulence and large-scale circulations? (ii) How do the

radiative effects of water vapor and clouds influence shallow circulations and convection? (iii) To what extent do mesoscale patterns of convective organization condition the response of clouds to perturbations? (iv) What are the implications of all of the above for how clouds respond to warming?

In order to respond to those questions, remote sensing and in situ observations on various scales were performed to investigate clouds and precipitation and their atmospheric and oceanic environment (Stevens et al., 2021). Four research vessels, four



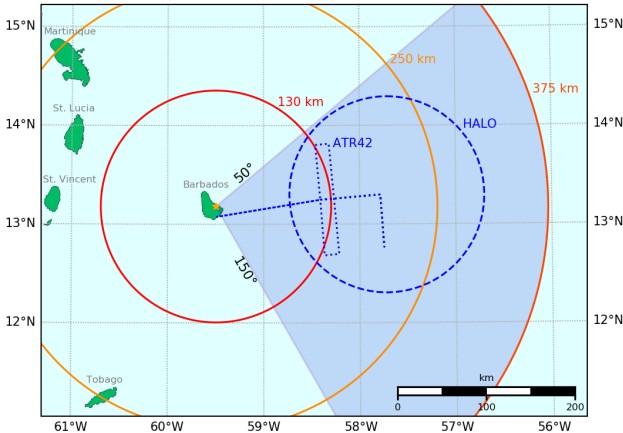

**Figure 1.** Coverage of the EUREC⁴A experimental area by Poldirad (blue shaded sector). Blue lines represent the typical flight pattern of the German HALO (dashed) and French ATR-42 (dotted) research aircraft.

research aircraft, a large number of autonomous boats and aircraft were involved. Additionally measurements were taken on the island of Barbados.

In this context the mission and the contribution of the dual-polarization Doppler weather radar Poldirad was:

  – to provide an overview for research aircraft and vessels in the experimental area east of Barbados (see Fig. 1),

  – to investigate the life cycle of the precipitation cells approaching Barbados and in particular those passing over the
Barbados Cloud Observatory (BCO Stevens et al., 2016),

  – to investigate the possibilities about cloud microphysics retrieval when combining C-band (wavelength 5.5 cm) measurements with profiles from the Ka-band (wavelength 8 mm) radar at BCO.

To support the first item, Poldirad images and data were transferred in real-time to Barbados Meteorological Service and incorporated into the Caribbean Radar Composite. Real-time images were available through the Atmosphere Planet software
system[1] onboard the German HALO (Hight Altitude and LOng range research aircraft) and French ATR-42 research aircraft, and for ground staff. Radar images were also of importance for the flight crew of the French remote controlled unmanned aerial vehicle *Boréal* which should avoid flying into precipitation.

## 3   Poldirad on Barbados

The C-band polarimetric weather radar Poldirad (Polarization Diversity Radar, Schroth et al. (1988), Fig. 2) was deployed on
Barbados for the EUREC⁴A campaign. Due to unforeseeable long delays during shipping from Germany, custom handling and local permissions, the radar was ready for operation on Feb. 5ᵗʰ with a two week delay for the campaign. Originally, it was

---

[1]http://www.atmosphere.aero/products-services/planet, last access: 27 April 2021.

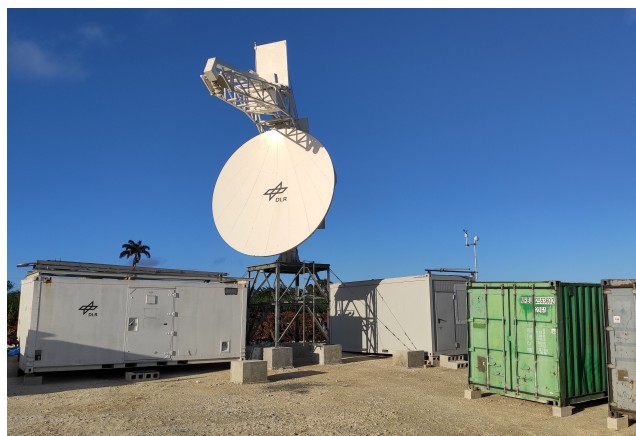

**Figure 2.** C-band radar Poldirad on Barbados (photo by F. Ewald).

planned to operate the radar after the campaign with a reduced schedule until June 2020. However, due to the rapid outbreak of the SARS-CoV-2 virus and the restriction in traveling, operation was stopped in March 2020. A limited radar operation was resumed mid May 2020 until a major failure of the antenna drive at the beginning of August 2020.

## 3.1 Radar system

Poldirad was installed in 1986 at the DLR (Deutsches Zentrum für Luft und Raumfahrt – German Aerospace Center) site Oberpfaffenhofen (Germany) as one of the first polarimetric weather radars in Europe and is presently – after some major upgrades – up-to-date (Table 1). Its unique fast ferrite polarization network allows for any kind of polarization basis – even elliptical ones. The polarization basis can be independent for transmit and receive and can be changed from pulse to pulse.

During EUREC⁴A two modes were used: (i) the hybrid or STAR (Simultaneous Transmit And Receive) mode and (ii) the alternate HV mode (Horizontal Vertical). In the hybrid mode a linear polarized pulse with an orientation of 45° is transmitted and on reception the signal is received simultaneously with linear horizontal and vertical polarization. In the alternate HV mode the polarization on transmit is alternating from pulse to pulse from linear horizontal to linear vertical polarization. On reception always both, the linear horizontal and vertical polarization are recorded. Only the later mode allows measuring the

full back-scattering matrix and hence the depolarization signal (LDR: Linear Depolarization Ratio).

In contrast to most other C-band weather radar systems, Poldirad is equipped with an offset antenna and the antenna is not sheltered by a radome. With an offset design the antenna feed does not cause blockage of the antenna beam and cannot create additional cross-polar signal making it better suitable for depolarization measurements. During the development of Poldirad it was decided to design the radar without a protective radome, since it was not appraisable how much a radome (especially a wet

one) will influence polarization purity.



For the mobile deployment of Poldirad the antenna can be installed on a 3 m high tower. Radar control electronics are housed in one 20 ft container while another 20 ft office container is used for radar operation (Fig. 2). Two further 20 ft containers are used for transportation of the equipment.

**Table 1.** Technical Specifications of Poldirad. Note the different settings for the long range and short range modes.

| Parameter | Value(s) | |
|---|---|---|
| Frequency | 5.504 GHz | |
| Wavelength | 5.45 cm | |
| Antenna diam. | ca. 4.5 m | |
| Beamwidth (half-power) | ca. 1.0° | |
| Modes | short range | long range |
| Peak transmit power at antenna feed | ca. 145 kW | ca. 185 kW |
| Pulse length | 1.0 µs | 2.0 µs |
| Pulse repetition frequency | 1150 Hz | 400 Hz |
| Maximum range | 130 km | 375 km |
| Range resolution | 150 m | 300 m |
| Polarization mode | alternate HV | hybrid |
| Sensitivity (dBz @ km) | −20@10, −5@50, 2@120 | −11@50, −4@120, 4@300 |

The sensitivity or lowest observable reflectivity, named minimum detectable signal (MDS) $Z_m$ is a function of range $r$ (cf. Fig. 4 (bottom) and Fig. 12)

$$Z_m(r) = Z_0 + 20 \log_{10} r + K_a r \tag{1}$$

with $Z_0$ as MDS in dBz at a range of 1 km, $r$ range in kilometres, and $K_a$ the two-way atmospheric attenuation in dB per kilometre. $Z_0$ is −47.5 dBz for the long range mode and −39 dBz for the short range mode. Poldirad's signal processor uses $K_a = 0.019$ dB km$^{-1}$ as an estimation of the two-way atmospheric attenuation at C-band. The actual sensitivity is slightly varying within ca. 0.5 dB since the transmit power is varying during the course of the scanning cycle in dependency of the transmitter temperature. Note that, due to internal settings in the signal processor, the lowest recorded reflectivity value is −31.5 dBz. This limitation is only relevant for the first 6.2 km in long-range mode and 2.4 km in short range mode, respectively.

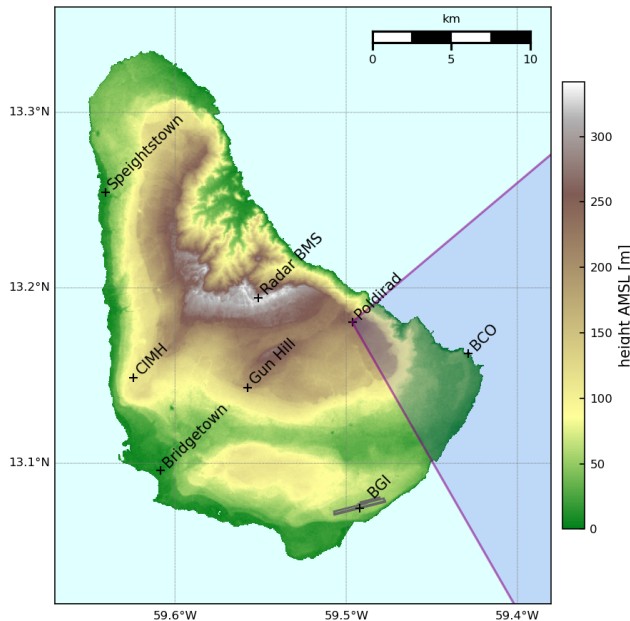

**Figure 3.** Location of Poldirad on Barbados. Shaded region shows scanning sector. BCO: Barbados Cloud Observatory, BGI: Barbados Grantley Adams International Airport, BMS Barbados Meteorological Service, CIMH: Caribbean Institute for Meteorology and Hydrology. SRTM 1 arcsec digital elevation model provided by U.S. Geological Survey.

## 3.2 Location

The main mission of Poldirad during EUREC$^4$A was the provision of precipitation information in the trade wind region east of Barbados. Therefore the radar location needed an unobscured view towards the experimental area. Logistical limitations were given by the facts (i) road access for trucks carrying four 20 ft sea containers; (ii) 3-phase power line nearby; (iii) no limitations by housing or vegetation; and (iv) internet access possible by land line or over the air.

A suitable location (59.49650° W, 13.18043° N, altitude approximate 240 m AMSL) was found near St. John's church,
St. John (Fig. 3) next to highway H. The site provides free view to the east, however, due to the village nearby and C-band installations near Barbados Grantley Adams International Airport (BGI) and next to Gunn-Hill signal station, radar scanning was limited towards the east sector.

## 3.3 Scan modus

The radar system is designed for research applications and allows highly flexible definition of scanning patterns. During the
campaign a 5 minute schedule with a long range surveillance PPI (Plan Position Indicator) scan in hybrid polarization mode up to 375 km at an elevation angle of 0.3°, a short range volume PPI scan in alternate HV polarization mode up to 130 km with the elevation angle sequence (0.0, 0.5, 1., 1.5, 2.0, 2.5, 3.0°), and a short range RHI (Range Height Indicator) scan in alternate

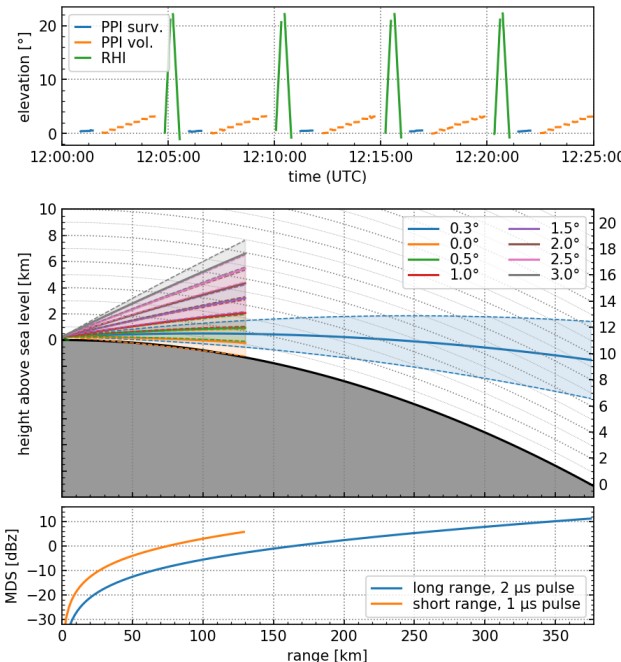

**Figure 4.** Top: sample of temporal sequence of long range surveillance PPI (blue), short range volume PPI (orange), and short range RHI towards BCO (green). Middle: beam propagation relative to sea surface for long range (blue) and short range PPIs. Shaded area represents 1° beamwidth. Bottom: minimum detectable signal for long and short range modes.

HV polarization mode up to 130 km were performed. The PPI scans were limited towards east between 50° and 150° from North.

RHI scans were towards BCO at Deebles Point with an azimuth of 105° (distance 7.6 km) and an additional RHI towards 100° to capture precipitation cells approaching from the east towards BCO. Occasionally, targeted RHI scans were performed to track individual moving cells or precipitation near research aircraft.

Figure 4 shows in the upper panel the temporal sequence of the three scans. Depending on various factors, the three scans can take more than 5 minutes. In this case, the scheduling algorithm omits the RHI scan from time to time. The gaps between the scans are used for positioning the antenna and for noise sampling. The middle panel shows the beam propagation relative to sea surface of the PPI scans in long and short range mode. The lower panel shows the minimum detectable signal (MDS) for the two range modes as given by Eq. (1).

### 3.4 Data quality

Several aspects are related to the quality of radar data: (i) calibration; (ii) antenna pointing direction; (iii) sanity checks; and (iv) polarization purity. Most important is the calibration of the radar system. An absolute calibration can only be done using a target with well known backscatter properties lifted into the air. Normally a metal sphere with a diameter which is large

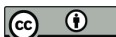



compared to the wavelength are used for that purpose. This is hardly achievable in the field. The architecture of Poldirad is
that the transmitter is located in the radar cabinet at ground whereas the polarization network and the receivers are mounted at
the rotating antenna. The transmit power is measured for each pulse by the receiver. During the deployment at Barbados the
signal path from the antenna feed horn to the receiver was not changed. For these reasons the calibration parameters determined
during the latest system upgrade are kept for the campaign. In order to account for temporal variations of the receiver chain,
a noise sample is taken before each scan. Additionally an offset calibration is performed using a single frequency signal with
defined power injected at the receiver frontend.

Verifying the antenna pointing direction and the alignment of the pedestal is frequently done by using the sun as radiative
source (e.g. Huuskonen and Holleman, 2007; Altube et al., 2015). The initial alignment was done with dedicated solar scans
(Reimann and Hagen, 2016, cf. Appendix A). Solar hits during the regular scan sequence were used to verify the antenna
pointing. Measurements after sun rise and as long as the sun was within the volume PPI or RHI scan revealed that elevation
error was less than $0.1°$ and that azimuth error was about $0.5°$ to the left.

The radar signal processor (Selex GDRX©) performs quality checks, ground clutter filtering, and second-trip echo removal.
Data with a low signal quality index (SQI) i.e. with too large standard deviation are disregarded and no further radar moments
are calculated. A ground clutter filter operating in frequency domain is applied by default to Poldirad radar parameters. Second-
trip echoes are considered if reflectivity is above noise level, but no phase coherence can be achieved within the sampling
interval.

The ferrite polarization network of Poldirad allows for a high flexibility in defining the polarization base for transmission
and reception (Schroth et al., 1988). However, the required settings for power division and phase delay need calibration.
This was done by Reimann (2013) and it was not possible to repeat this procedures in the field. A simple way to check the
polarization configuration is the assumption that for light rain the differential reflectivity $Z_{DR}$ should be close to zero (e.g.
Gourley et al., 2006) and LDR much below $-30$ dB. Occasionally setting the polarization fails, which becomes visible in data
where reflectivity is much less compared to the previous or following scan. These data should be omitted.

During the warm-up phase of the magnetron (about one hour after switching on the transmitter), the nominal frequency is
not yet reached and the receiver may fail to lock to the STALO[2] frequency. This occurs mainly for the long range PPI scans
and becomes visible in the data when reflectivity and other products are empty and the raw data files are much smaller than the
others. These data should not be used for further processing.

### 3.5 Sea clutter identification

The backscatter from sea surface can considerably contaminate radar observations if a weather radar is located close to the
coast. Reflectivity from sea clutter was observed up to 35 dBz for Poldirad scans at $0°$ elevation and up to 15 dBz for $0.5°$
elevation (Fig. 5 top). Ground clutter from land surface is classically removed using filters for stationary targets. However,
since sea surface is in motion, ground clutter filters will fail and additional filters have to be applied.

---

[2]stabilized local oscillator



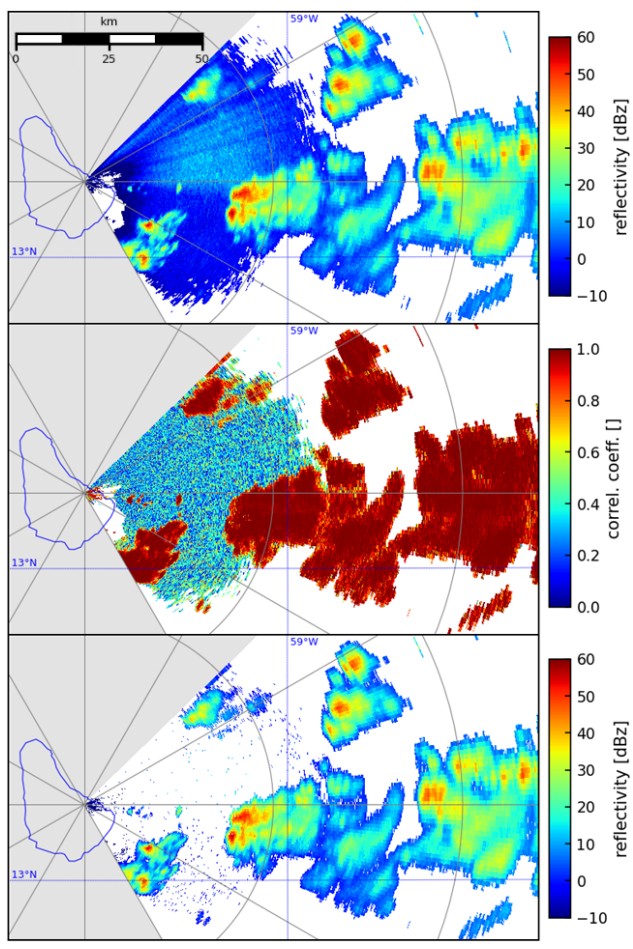

**Figure 5.** PPI scan with elevation $0.5°$ on Feb. $11^{\text{th}}$, 2020 at 12:07:25 UTC. Top panel: reflectivity horizontal polarization $Z_H$. Middle: copolar correlation coefficient $\rho_{HV}$. Bottom: reflectivity $Z_H$ where $\rho_{HV} > 0.7$.

Various algorithms do exist to identify hydrometeors using polarimetric radar data (e.g. Vivekanandan et al., 1999; Park
et al., 2009). The separation of meteorological targets from non-meteorological targets is a further aspect of some classification
techniques (e.g. Giuli et al., 1991; Berenguer et al., 2006), and more recently by Kilambi et al. (2018) or Overeem et al.
(2020). Besides some other properties observed in polarimetric parameters, like the texture of differential reflectivity $Z_{DR}$ or
differential propagation phase $\phi_{DP}$, the copolar correlation coefficient $\rho_{HV}$ is one of the most promising parameter to identify
sea clutter. Meteorological echoes have a high correlation between reflectivities from horizontal and vertical polarization.
Ryzhkov et al. (2002) proposed to use a threshold of $\rho_{HV} = 0.7$ to discriminate between meteorological echoes and sea
clutter. Figure 5 shows the observed reflectivity factor, copolar correlation coefficient and the filtered reflectivity factor using
a threshold of 0.7 for $\rho_{HV}$. Some speckles are not removed and also some weak echoes at the edges of precipitation cells



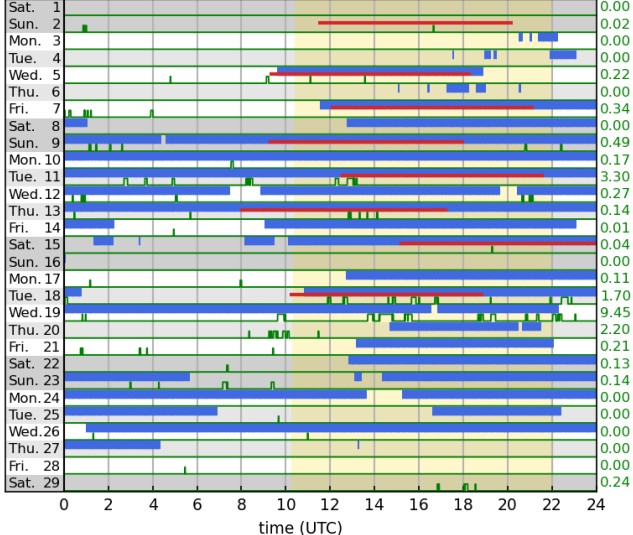

**Figure 6.** Availability of Poldirad data during February 2020 (blue bars) with rain events at BCO (green ticks) and HALO flight times (red lines). Green numbers to the right give daily precipitation sums (in mm) at BCO. Yellow shaded area indicates daylight time at Barbados.

are removed. The latter is caused by non-uniform beam filling while scanning across the edges of precipitation. For further processing of Poldirad data this threshold will be applied.

## 3.6 Availability of Poldirad measurements

Figure 6 shows the radar operation during February 2020 (blue bars) together with rain events (any non-zero 10 second rain intensity) recorded at BCO (green ticks) and HALO flight times (red). The flight on Feb. 18[th] was the return flight to Germany. The radar is designed for research application and is not suited for unattended 24/7 operation. During the campaign a 24-hour operation was envisaged. Failures and limited personal resources caused gaps in a continuous operation during EUREC[4]A and afterwards.

Further measurements, only during daytime, were performed from mid-May until beginning of August. During the nearby passage of the Tropical Storms Gonzalo (July 23[rd] to 25[th]) and Isaias (July 28[th] to 29[th]), Poldirad was operated continuously also during nighttime.

## 3.7 Data format

Poldirad raw data are stored in HDF5 (Hierarchical Data Format version 5) format following the ODIM 2.0.1 specifications (Michelson et al., 2010). This data structure was developed by the weather radar operators group of European meteorological services to ease the international exchange of weather radar data. Some additions and modifications to the ODIM specification had been necessary, since Poldirad offers more flexibility and options than operational weather radar systems. Each surveillance PPI, volume PPI, or RHI scan is stored as one file with an approximate size of 2 to 12 MBytes. Each file consists of general





meta data, meta data for each sweep or sub-scan (named as `dataset1` to `datasetn`) as well as for each radar product (named as `data1` to `datan`). Details on the structure and the attributes (meta data) can be found in Michelson et al. (2010). Depending on the scanning mode up to 22 radar products are defined. The available radar products are listed in Appendix B.

### 3.8    Gridded dataset

To ease the usage of Poldirad data within the EUREC⁴A community, a 2-dimensional gridded dataset was generated from the

long range surveillance scans. The sector scans were interpolated on a 1 by 1 km grid with a size of 400x400 km². Interpolation was done using the `griddata` routine from the Python SciPy package. For each grid point reflectivity factor $Z$, rainfall rate $R$, longitudinal ($x$) distance from radar, meridional ($y$) distance from radar, longitude, latitude, and approximate height above sea level is provided. Sea clutter was removed according to Sect. 3.5.

    Conversion from reflectivity to rainfall rate was done by the commonly used empirical $z$–$R$ relation $z = 200R^{1.6}$ (Marshall

et al., 1955) with $R$ in mm per hour and $z$ in $\text{mm}^6\,\text{m}^{-3}$. There are only a few $z$–$R$ relations attributed to shallow trade wind showers, e.g. Stout and Mueller (1968) ($z = 126R^{1.47}$) or Snodgrass et al. (2009) ($z = 88R^{1.6}$). These empirical relations would give about twice the rain rate for the same reflectivity. However, they are based on small sample sizes and thus might not be representative for the events observed during EUREC⁴A.

## 4    Examples from Poldirad measurements

In this section we will show some exemplary measurements from the Poldirad observations during the EUREC⁴A campaign.

### 4.1    Rain cell patterns

One of the main objectives of the EUREC⁴A campaign is to investigate the mesoscale cloud patterns and the life cycle of shallow convection in the trade wind zone east of Barbados. According to Stevens et al. (2020) and Bony et al. (2020) *Sugar*, *Fish*, *Gravel*, and *Flowers* like patterns can be classified from satellite images. Precipitation patterns observed by Poldirad are

related to these cloud patterns.

    Figure 7 (top) shows a maximum display from 5 Feb. 2020 at 12:03 UTC. This kind of displays gives the horizontal distribution of precipitation as well as the vertical extent of the cells. The two side views are the maximum projection through the 3-dimensional volume from south to north and from west to east, respectively. Some speckles of not completely removed sea clutter are visible with reflectivity values below ca. $-5$ dBz. That day was classified by the EUREC⁴A community as

being dominated by *Gravel* type cloud patterns. The size of the cells are in order of 15 km, their height is 3 to 4 km and the distribution is random with some clustering.

    On 11 Feb. 2020 rain cells were larger and widespread (Fig. 7 bottom) and did reach heights of 4 to 5 km. Cells had a multi-cell structure with multiple cores of high precipitation. Cloud patterns on that day were classified as *Flowers*. With the limited view of the volume scan the whole *Flowers* pattern is not visible.

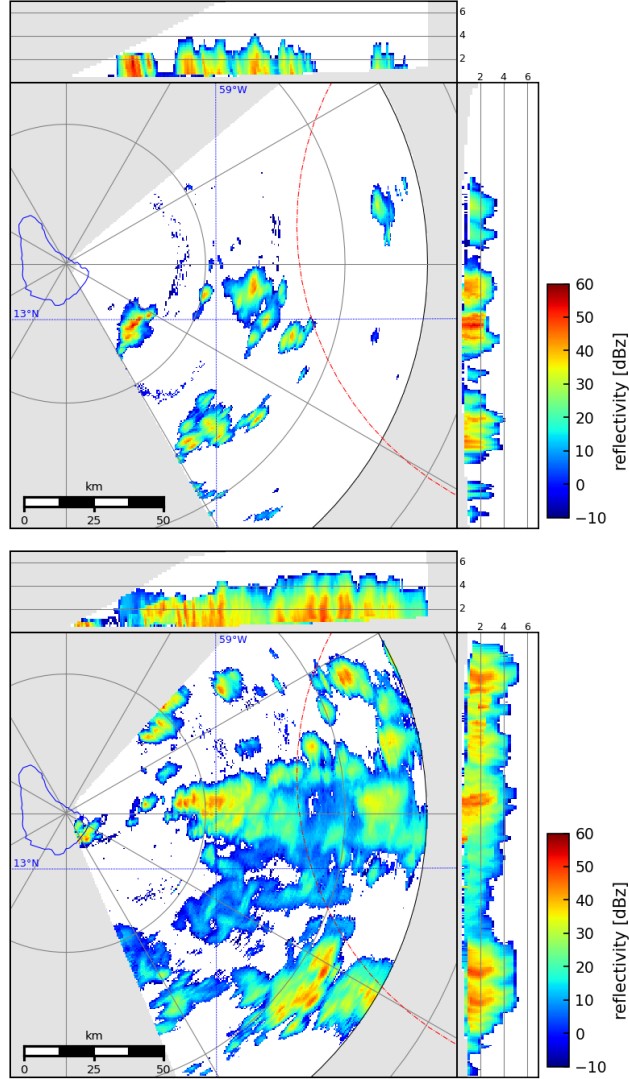

**Figure 7.** Maximum display (details see text) of reflectivity. Red dot-dashed line shows HALO circle pattern. Heights of side projections are in km. Top: 5 Feb. 2020, 12:03 UTC; bottom: 11 Feb. 2020 07:27 UTC.

## 4.2 Daily accumulated precipitation and daily cycle

With long range surveillance scans about every 5 minutes daily rainfall accumulations can be generated and e.g. the diurnal cycle of precipitation can be studied.

Figure 8 (top) shows the daily accumulated rainfall pattern for Feb. 9[th] 2020 derived from the gridded datasets. The average daily precipitation amount within the HALO circle was 0.18 mm. Traces of individual cells can be located throughout the day. Beyond approximate 250 km range less cells are observed. Only a few cells were observed further out. This can be attributed



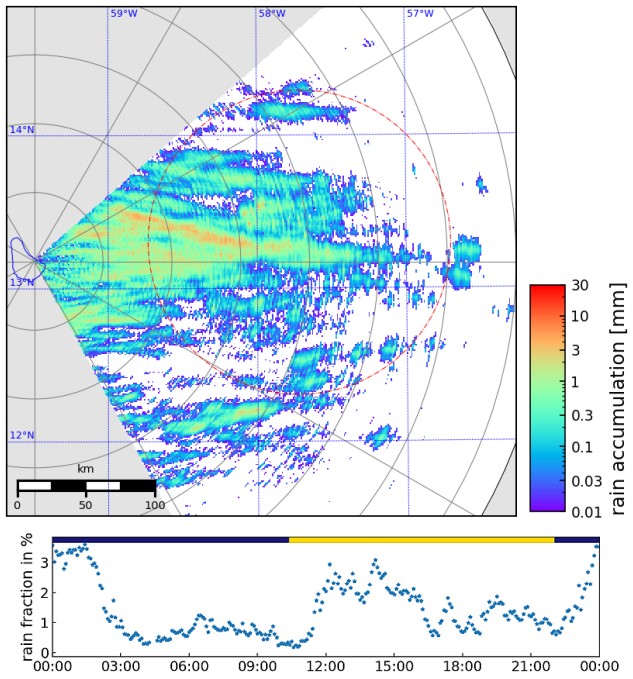

**Figure 8.** 9 Feb. 2020. Top: daily rain accumulation, red circle denotes the HALO circle pattern. Bottom: percentage of precipitation area within the HALO circle pattern, times are in UTC. Blue/yellow bar on top indicates night/day at Barbados.

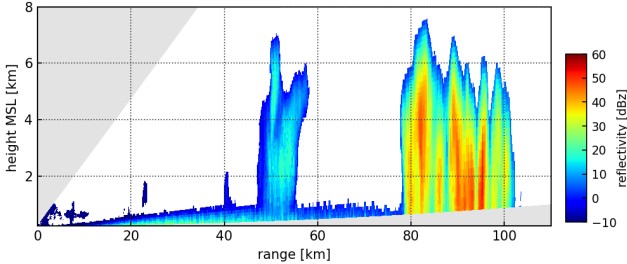

**Figure 9.** RHI scan on 9 Feb. 2020 at 16:19 UTC towards 71°. Note, shallow layer with low reflectivity is sea clutter.

to the vertical extent of the majority of the cells and height of the radar beam in dependence with range (cf. Fig. 4 center). In general, stronger precipitation is observed closer to the radar where the radar beam intersects within the more intense part of the precipitation cores of the cells (see Fig 9). In Fig. 8 (bottom) the diurnal cycle of precipitation area within the HALO circle flight pattern (red circle in Fig. 8 top) is shown. For that day, only a weak diurnal cycle of precipitation can be identified.

On Feb. 11th the situation was different (Fig. 10): more precipitation was observed in the area. The average daily precipitation amount was 0.95 mm within the HALO circle. The cells were more widespread and more frequent. Vertical extent of the precipitation was up to approximate 4 km (Fig. 11) and lower thus than on Feb. 5th. The rain fraction within the HALO circle





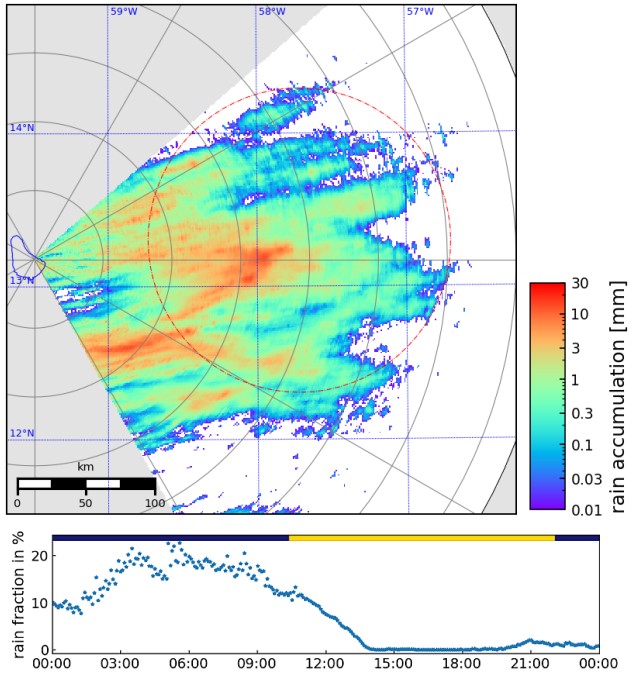

**Figure 10.** As Fig. 8 for 11 Feb. 2020.

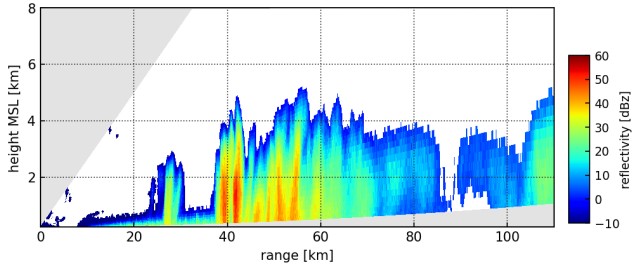

**Figure 11.** RHI scan on 11 Feb. 2020 at 07:30 UTC towards $86°$. Note, shallow layer with low reflectivity is sea clutter.

shows a pronounced daily cycle. Up to 20 % were observed in the morning hours and almost no precipitation during daytime. This agrees well with the simulations and long-term observations by Vial et al. (2019).

### 4.3 Long-range reflectivity observations

As shown above, long-range reflectivity observations are limited by a number of factors: (i) earth-curvature and the height of the radar beam above ground, (ii) broadening of the radar beam and loss of horizontal and vertical resolution, and (iii) the limitations by the minimum detectable signal (MDS). In Fig. 12 we show the frequency of observed reflectivity values with range for all long-range measurements on 11 Feb. 2020. The red line indicates MDS according to Eq. (1) with $Z_0 = -47.5$ dBz.





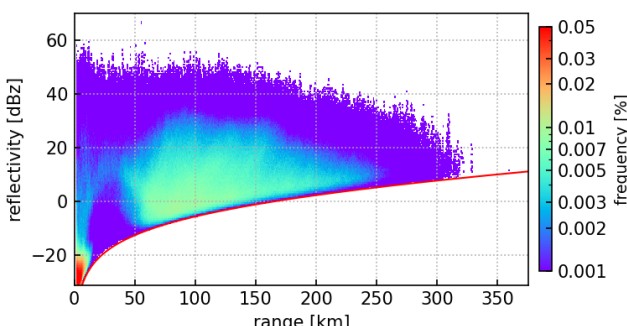

**Figure 12.** Reflectivity distribution with range for 11 Feb. 2020 from long-range measurements. Red line indicates minimum detectable signal (MDS) as given by Eq. (1).

Sea clutter has been removed. Frequent low reflectivity values ($Z < -20$ dBz) at short ranges indicate non precipitating cloud observations. The notch in the range from ca. 10 to 50 km is attributed to the masking of low level meteorological radar echoes by sea clutter. The frequency of echoes with reflectivity above 40 dBz stays fairly constant up to about 150 km, indicating that probably most of the high reflectivity precipitation cores can be observed up to that range. At further ranges the radar beam is only partial filled with echoes since only the upper part of the cells reach partly into the radar beam. This results in reduced

reflectivity values since the radar equation always assumes a completely filled measurement volume. For that day no more precipitation echoes were recorded beyond 320 km.

## 4.4    Vertical reflectivity distribution

Contoured frequency by altitude diagrams (CFAD) are often used to investigate vertical structures of radar observables from precipitation systems (Yuter and Houze, 1995) or to compare measurements with numerical simulations. Here we use the fre-

quent RHI scans towards BCO or other directions to derive reflectivity CFADs. In contrast to CFADs derived from vertical pointing radars, we have to consider the range dependent minimum detectable signal, as well as the inability to take measurements close to the surface at far ranges, the limited altitude range of RHI scans for short ranges, and the broadening of the radar beam with range (cf. Fig. 4).

         Figure 13 shows CFADs for 11 and 15 February 2020. For the above mentioned reasons, CFADs are split into three range

sets, namely 0–20 km, 20–60 km, and 60–130 km. Only at the first range interval, shallow cumulus clouds with reflectivity values below $-10$ dBz are observed on both days. At longer ranges mainly rain showers are observed on February 11[th]. As also seen in Fig. 11 cells did to not exceed 5 km in altitude.

         On February 15[th] elevated moisture transport (Villiger et al., 2021) lead to a widespread stratiform cloud layer in the altitude range 4 to 9 km (Fig. 14). The cloud layer is visible in all three CFAD range intervals. Low level clouds are visible mainly in the

short range interval. Beam broadening at far ranges overestimate the vertical extent in both, the RHI and CFADs, respectively. Figure 15 shows the CFAD from the HALO cloud radar (Mech et al., 2014; Ewald et al., 2019; Konow et al., 2021) while HALO was flying on its standard circle (Fig. 1) on an altitude of 11.2 km between 21:30 and 23:30 UTC. The airborne cloud



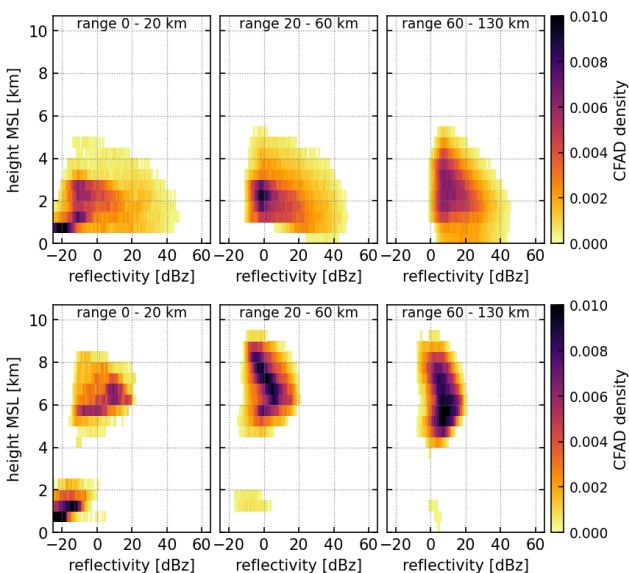

**Figure 13.** CFADs for 11 Feb. (top) and 15 Feb. 2020 (bottom). Range intervals are 0–20, 20–60, and 60–130 km.

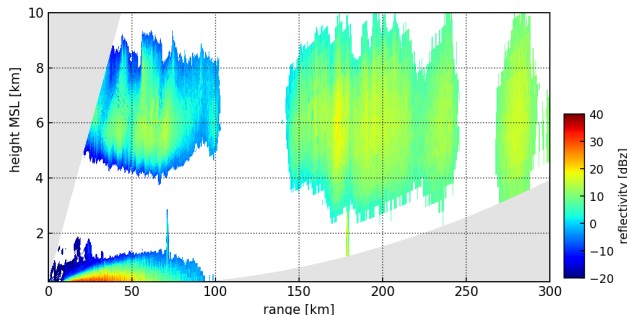

**Figure 14.** RHI scan on 15 Feb. 2020 at 22:40 UTC towards 105°. Note, shallow layer up to 95 km in range is sea clutter.

radar has a high vertical resolution (ca. 31 m) and a much higher sensitivity (ca. −38 dBz at an range of 5 km) compared to Poldirad (ca. 2 dBz at an range of 120 km). The structure of the CFADs show good agreement in the vertical extent of the

cloud layer. The agreement in the maximum observed reflectivity (ca. 20 dBz) confirms in a way the calibration of Poldirad since the HALO radar has been calibrated by independent means (Ewald et al., 2019).

## 5   Data availability

All data are currently considered as preliminary since calibration and data quality are based on best knowledge (cf. Sect. 3.4) and have not been validated so far. Raw data are available on the EUREC⁴A AERIS database via digital object identifier (DOI)





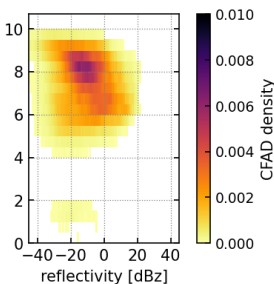

**Figure 15.** CFAD from HALO cloud radar for 15 Feb., 21:30 - 23:30 UTC.

https://doi.org/10.25326/218 (Hagen et al., 2021a). Gridded data (cf. Sect. 3.8) are available via https://doi.org/10.25326/217 (Hagen et al., 2021b).

## 5.1    Poldirad level 1 raw data

Poldirad raw (level 1) data are stored in HDF5 format following the ODIM 2.0.1 specifications (Michelson et al., 2010). Additional details are given in Sect. 3.7. The data are as they have been generated in real-time by the radar signal processor. A
description of the radar parameters is given in Appendix B.

Data folders are named according to the type and object of the scan:

`PPI_PCP/`: PPI long range surveillance precipitation scan

`PPI_VOL/`: PPI short range volume scan

`RHI_BCO/`: RHI short range scan towards BCO

`RHI_TAR/`: RHI scan towards moving target like precipitation cells or aircraft

Files are stored in daily directories `YYYY/YYYY-MM-DD` with `YYYY` the year (2020), `MM` the month, and `DD` the day.

Files are named according to date (YYYYMMDD), time (hours `HH`, minutes `MM`, seconds `SS`), type and object as defined above, angles, range, and a reference number:

`POLDIRAD_TYPE_OBJECT_YYYMMDD_HHMMSS_ STARTANG_STOPANG_MAXR_REFNUM_ preliminary.hdf5`
with:

`STARTANG` / `STOPANG`: start and stop angles:

for PPI: start/stop elevation angle in tenths of degrees

for RHI: start/stop azimuth angle in degrees

`MAXR`: maximum scan range in km

`REFNUM`: Internal reference number to raw data directory and file structure (`SSSIIII`):

`SSS`: storm number and `IIII`: scan number



## 5.2 Poldirad level 2 gridded data

The gridded (level 2) data are retrieved as described in Sect. 3.8 from the `PPI_PCP` dataset and stored in NetCDF format with the same file name convention.

Two versions are currently available:

`/PPI_PCP_L2GRID/`: version 0

`/PPI_PCP_L2GRID_v1/`: version 1

Version 1 has in addition to the data fields of version 0 longitude and latitude of the grid points and the height of the radar beam above sea level.

Quicklooks of gridded rain rate are available

`/PPI_PCP_L2GRID_QL/`: quicklooks version 0

`/PPI_PCP_L2GRID_QL_v1/`: quicklooks version 1

## 6  Conclusions

After some initial delay during the first two weeks of the EUREC[4]A campaign, Poldirad was able to capture various precipita-
tion systems during the remainder of the field campaign (starting Feb. 5[th], 2020) and during summer 2020. Radar images have been used for flight planning through real-time image transfer (see Fig. 16) which were available onboard research aircraft and for campaign participants on ground. Additionally, the radar images were incorporated in real-time into the Caribbean Radar Mosaic.

Data processing and quality control is performed by the implemented receiver and signal processor. Using dual-polarization
correlation coefficient $\rho_{HV}$ with a threshold of 0.7 proved to be a suited parameter to eliminate contamination of radar data by sea clutter. A preliminary comparison with observations from the HALO cloud radar shows that the calibration of Poldirad is reliable, details will be subject for further detailed studies. Processed data are available as HDF5 files (Hagen et al., 2021a), as well as gridded reflectivity and rain rate fields (Hagen et al., 2021b).

Poldirad adds a valuable contribution to the multifaceted data gathered during the EUREC[4]A campaign. Preliminary analyses
of reflectivity distribution show that Poldirad could capture rain cells up to a range of 250 to 300 km. The typical vertical extent of shallow rain cells is in the range of 3 to 6 km, their size is in the order of 10 to 20 km. On some days a pronounced diurnal cycle of precipitation was observed. The high temporal and spacial frequency of long range and volume PPI scans will allow for ongoing and further detailed studies of the life cycle of shallow cumulus clouds in the trade wind region east of Barbados.

## Appendix A:  Measurement of antenna pattern

For transportation Poldirad's antenna has to be disassembled into four panels (faintly visible in Fig. 2). Reassembling is performed with high accurateness on the ground before lifting the antenna to the pedestal. Even though the supporting structure is very rigid, it can not be guaranteed, that the antenna will be mounted with the very same shape like it had on a factory antenna

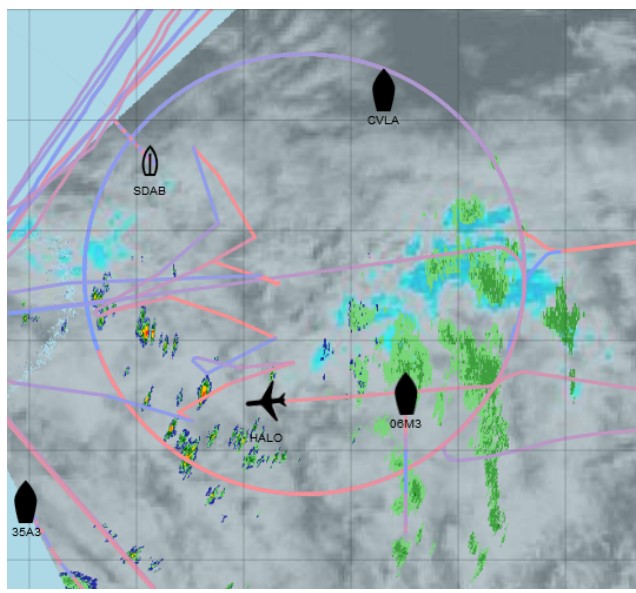

**Figure 16.** Screenshot example of the Planet software display for 15 Feb. 2020 at 23:48 UTC showing Poldirad image (colors from green to red), GOES-East infrared image (gray and light blue), tracks from research vessels, and the flight tracks from German HALO (circle pattern and excursions) and British Twin-Otter (zigzag-like pattern) as well as commercial air traffic in and out of Barbados (straight diagonal lines on the upper and lower left side).

range in December 1984. The shape of the parabolic reflector defines the shape of the radar beam. Initial measurements show a nearly circular beam pattern with a half-power beamwidth of approximately 1° (see Fig. 3 in Schroth et al., 1988). Detailed

measurements of the shape of the radar beam were performed in 2012 by Reimann and Hagen (2016) using the sun and an external signal source.

Dedicated solar scans were repeated after the set up of the radar on Barbados. Figure A1 shows the measurements on Feb. 3$^{rd}$, 2020 similar to Fig. 4 in Reimann and Hagen (2016). The top image shows the individual measurement points during the scan, azimuth and elevation angles have been corrected for high elevation angles as described by Reimann and Hagen (2016).

Maximum received power was −103.2 dBm. Using a two-dimensional least-square fit to Gaussian shape of the main lobe (Huuskonen and Holleman, 2007) gives a half-power beamwidth of 1.2° in azimuth and 1.0° in elevation (black ellipses). The pointing offset is 0.45° in azimuth and 0.17° in elevation, respectively (black star). The light blue lines indicate the −3 and −10 dB power isolines from the peak of the measured power. While the central part of the radar beam is nearly circular, the outer part has an elliptical shape which is much wider in azimuth. The reasons causing this deformation are not clear. The two

notches at the lower side are probably caused by the boom carrying the feed horn. The two lower images show a horizontal along azimuth (left) and vertical along elevation cut (right) at the beam center through the measurements (gray points). Red points are within ±0.1° of the beam center. Black line is the shape from the least-square fit. Dotted blue lines indicate the maximum and −3 dB power as well as the beamwidth and the offset. It becomes visible that the beam shape is following the

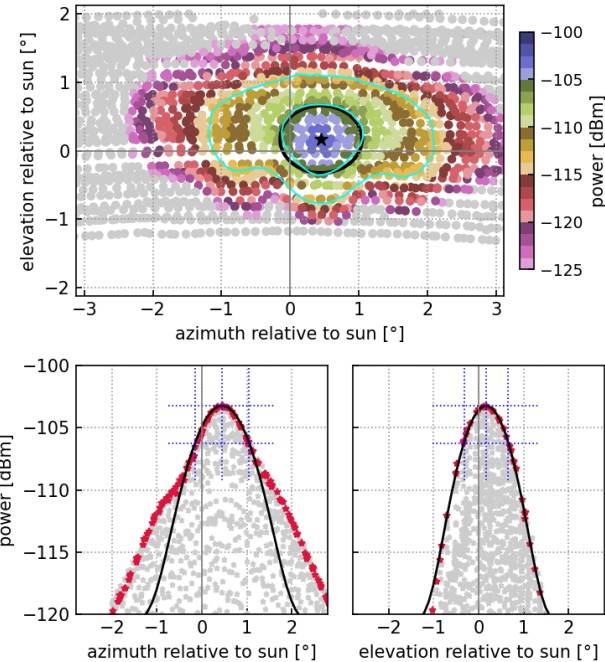

**Figure A1.** Power received from the sun on 3 Feb. 2020 from 13:37 until 14:29 UTC. Average sun azimuth $132°$, elevation $46°$. Top: received power, light blue lines indicate $-3$ and $-10$ dB power isolines, black line $-3$ dB power isoline from Gaussian fit. Bottom: left azimuth, right elevation cut, red dots are within $\pm0.1°$ of the beam center. Further details see text.

Gaussian approximation in elevation while in azimuth a considerable deviation starts about $-5$ dB below the maximum power
level.

## Appendix B: Radar products

Table B1 gives a short description of radar product identifiers used in the HDF5 files. The two different polarization modes require different signal processing and thus different radar products are generated. Physical details of radar products and their interpretation can be found in text books like Bringi and Chandrasekar (2001).

*Author contributions.* MH, FE, and SG are the lead investigators for the deployment, LO is caring for all technical aspects, DF and MF provided all administrative, logistical, and scientific support on the island of Barbados. MG, JH, JR, EM, and ET supported the technical installation, JR additionally cared for the radar software refinements. GM, CK, QL, RCL, AD, DG, and KH additionally supported the measurements during EUREC[4]A and after the campaign.



*Competing interests.* The authors declare that they have no conflict of interest.

*Acknowledgements.* We are deeply grateful to the Max Planck Institute for Meteorology, Hamburg, Germany, namely Bjorn Stevens, Lutz Hirsch, and Friedhelm Jansen for logistic support and gathering considerable financial funding by the Supporting Members of the Max Planck Society for the deployment of Poldirad at Barbados. We thank Daison Lowe and Ryan White from CIMH for their technical assistance. Samuel Estwick (Sammy) cared sedulous for electricity at the site. A large number of EUREC$^4$A participants helped us setting up the radar dish and tower on the "barn rising day", January 25$^{th}$, 2020. Barbados Meteorological Service established internet connection during the
installation phase and initiated the data transfer to the Caribbean Radar Composite.





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





**Table B1.** Poldirad radar products and their HDF5 identifiers. Correlation coefficients and signal quality indices are dimensionless.

| HDF5 identifier | alt-HV mode | hybrid mode | product | unit | description |
|---|---|---|---|---|---|
| DBZH | x | x | $Z$, $Z_H$, or $Z_{HH}$ | dBz | horizontally polarized corrected reflectivity factor |
| DBZV | | x | $Z_V$ or $Z_{VV}$ | dBz | vertically polarized corrected reflectivity factor |
| TH | x | x | $T_H$ or $T_{HH}$ | dBz | horizontally polarized uncorrected reflectivity factor |
| TV | x | x | $T_V$ or $T_{VV}$ | dBz | vertically polarized uncorrected reflectivity factor |
| TVH | x | | $T_{VH}$ | dBz | cross-polar vertically polarized uncorrected reflectivity factor at horizontally polarized transmitted pulse |
| THV | x | | $T_{HV}$ | dBz | cross-polar horizontally polarized uncorrected reflectivity factor at vertically polarized transmitted pulse |
| ZDR | x | x | $Z_{DR}$ | dB | differential reflectivity ($Z_H - Z_V$) |
| LDR | x | | LDR | dB | linear depolarization ratio ($Z_{VH} - Z_H$) |
| LDRV | x | | $\text{LDR}_V$ | dB | linear depolarization ratio ($Z_{HV} - Z_V$) |
| RHOHV | x | x | $\rho_{HV}$ | - | copoloar correlation coefficient |
| PHIDP | x | x | $\phi_{DP}$ | degrees | differential propagation phase |
| KDP | x | x | $K_{dp}$ | degrees km$^{-1}$ | specific differential propagation phase |
| SQI | x | x | SQI | - | signal quality index from horizontally polarized samples |
| SQIV | x | x | $\text{SQI}_V$ | - | signal quality index from vertically polarized samples |
| SQIHV | x | | $\text{SQI}_{HV}$ | - | cross-polar signal quality index from vertically polarized samples at horizontally polarized transmitted pulse |
| VRAD | x | x | $v$ | m s$^{-1}$ | alt-HV mode: radial velocity combined from horizontally and vertically polarized samples |
| | | | | | hybrid mode: radial velocity from horizontally polarized samples |
| VRADH | x | | $v_H$ | m s$^{-1}$ | radial velocity from horizontally polarized samples only |
| VRADV | | x | $v_V$ | m s$^{-1}$ | radial velocity from vertically polarized samples |
| WRAD | x | x | $w$ or $\sigma_v$ | m s$^{-1}$ | alt-HV mode: spectral width of radial velocity from horizontally and vertically polarized samples |
| | | | | | hybrid mode: spectral width of radial velocity from horizontally polarized samples |
| WRADV | | x | $w_V$ or $\sigma_{vV}$ | m s$^{-1}$ | spectral width of radial velocity from vertically polarized samples |
| CCOR | | x | CC | dB | clutter correction for horizontally polarized reflectivity factor |
| CCORV | x | x | $\text{CC}_V$ | dB | clutter correction for vertically polarized reflectivity factor |
| RHOXH_ABS | x | | $|\rho_{cxH}|$ | - | absolute value of horizontally polarized cross-polar correlation coefficient |
| RHOXH_ARG | x | | $\arg(\rho_{cxH})$ | degrees | argument of horizontally polarized cross-polar correlation coefficient |
| RHOXV_ABS | x | | $|\rho_{cxV}|$ | - | absolute value of vertically polarized cross-polar correlation coefficient |
| RHOXV_ARG | x | | $\arg(\rho_{cxV})$ | degrees | argument of vertically polarized cross-polar correlation coefficient |