# Peer review of "Deployment of the C-band radar Poldirad on Barbados during EUREC4A"

_Earth System Science Data, 2021_

## Author Comment (AC1)

**Authors' Response**

We would like to thank the reviewers for their comments on the manuscript. Their suggestions and recommendations will certainly help improving the readability of the manuscript.

In the following, we will answer the comments of both reviewers in one document. In blue we repeat the reviewer's comments, significant changes to the manuscript will be in red.

**Comments from Reviewer #1, Alan Blyth**

This paper describes the observations that were gathered by the Poldirad radar mainly during the EUREC4A field campaign. The paper is very clear, well written and the figures are well chosen. The paper provides a very useful summary of the Poldirad radar and the operations during EUREC4A and good description of the data files.

Thank you for this positive statement.

Line 31. Change "In this view" to "With this in mind"?

Corrected as suggested.

Line 32. "The non-polarimetric..."

Corrected as suggested.

Line 79. Latter

Corrected as suggested.

Line 90. I think reference to Fig. 4 comes before Fig. 3.

We have introduced a new figure 3 which shows now MDS only, MDS has been removed from figure 4 (is now 5)

Fig 4. Caption. It would be worth adding (MDS) even though it's obvious.

Corrected as suggested in the new figure 3.

Line 163. Parameters.

Corrected as suggested.

Line 193. "according to the procedure outlined in Sect. 3.5"?

Corrected as suggested.

Line 197. "... twice the rain rate for the same reflectivity..." is a bit ambiguous.

Changed to: When using these relations, the estimated rainfall rate would be about twice as high as when using $z = 200\,R^{1.6}$.

Line 210. Should "cells" be "rain cells" throughout? But then does that depend on the value of Z? Sometimes "cloud system" seems more appropriate.

This is a good point. We changed the heading of this section to "Precipitation patterns" to have a more general description, even though solid precipitation was probably rare during EUREC4A. We added a sentence to point to the difference between clouds and rain (or precipitation) in this context:

Due to the wavelength, clouds can be observed by Poldirad only at very close range. At far range only precipitation can be observed. Therefore, the term precipitation or rain patterns or cells will be used in the following when describing radar echoes.

Line 212. Clouds had a multi-cell structure?

I wonder if "high precipitation" should be "heavier precipitation" so there is no ambiguity with altitude?

Changed to: The precipitation patterns had a structure with multiple cores of heavier precipitation.

Line 214. Flower pattern.

Corrected as suggested.

Line 222, Figure 9. It is interesting to see an example of such deep clouds that will have contained ice and graupel particles during EUREC4A. (Just a comment!)

Occasionally we observed deep clouds, maybe it would be worth to make some statics. However, there are limitations by the scanning strategy with a focus on shallow clouds.

Line 227. Approximately. And thus lower?

Corrected as suggested.

Line 235. Is it possible that there was a very small concentration of precipitation-sized drops?

It can not be excluded and occasionally fall streaks had been seen by us. But, we think that reflectivities below -20 dBz are typically for clouds without larger (i.e. falling) particles.

Line 239, Figure 12. Would it be helpful to somehow include the information in the middle diagram of Figure 4 in Fig. 12?

The height of the radar beam has now been included.

Figure 13. Would it be useful to include the altitude of the 0 deg level in the bottom diagram?

We included the height of the 0°C isotherm in all CFAD diagrams. The height of the 0°C isotherm was derived from the soundings at BCO on those days (Stephan et al., 2021).

Lines 311-312. Was the diurnal cycle observed on several days? Is that the meaning of the full sentence on line 228?

Text changed to: On Feb. 11[th] a pronounced diurnal cycle as described by Vial et al. (2019) of precipitation was observed. On other days either no or only a weak diurnal cycle was observed, or the radar observations didn't cover the full day (cf. Fig. 7).

Line 326. Where are the black ellipses in Fig A1? There is one ellipse in the top diagram.

Text changed to: black lines in Fig. A1

Line 329. Is there any information at all on the possible effect of the deformation on the observations?

A new paragraph with Figure A2 and Table A1 have been added to discuss the effect of the deformation:

While the central part of the radar beam is nearly circular, the outer part has an elliptical shape which is much wider in azimuth. The reasons causing this deformation are not clear. The implication of the deformation is that observed precipitation patterns appear to be stretched in azimuthal direction. Figure A2 shows a simulated circular rain cell centered at a range of 100 km and an azimuth of 60° which is observed by a radar. Only the main lobe, no sidelobes have been considered in the radar simulation. Range effects, i.e. non-rectangular pulse shape are ignored. The left image shows the "true" reflectivity pattern, the central image shows the broadening as it would be observed by a radar having an azimuthal beamwidth of 1° (like most weather radars). The right image shows a gaussian approximation [footnote: central part with beamwidth 1.2°, outer part with beamwidth 2.3° with an offset of -4dB] of the Poldirad beam (c.f. Fig. A1 lower left).

The consequence of the broad radar beam is that the size of rain cells will be overestimated, while their maximum rain rate will be underestimated. Table A1 gives an estimation for the effect on maximum reflectivity, maximum rain rate, area where reflectivity is greater 7 dBz (rain rate greater 0.1 mm h$^{-1}$), and mean rain rate of the cell.

[Figure]

**Fig. A2.** Simulated effect of broadening of a circular rain cell when observed by a radar. Left: simulated rain cell; center: observed with a radar with beamwidth 1°; right: as center image, but with Poldirad beam. Range rings every 5 km, radials every 2°.

**Table A1.** Effect of azimuthal beamwidth on a simulated rain cell.

|  | true rain cell | beamwidth 1° | Poldirad beam |
|---|---|---|---|
| maximum reflect. | 50.1 dBz | 48.3 dBz | 47.6 dBz |
| max. rain rate | 49.0 mm h$^{-1}$ | 37.8 mm h$^{-1}$ | 34.3 mm h$^{-1}$ |
| area ($Z > 7$ dBz) | 13.1 km² | 19.1 km² | 29.4 km² |
| mean rain rate | 7.9 mm h$^{-1}$ | 6.4 mm h$^{-1}$ | 4.6 mm h$^{-1}$ |

Corrected as suggested.

Corrected as suggested.

Corrected as suggested.

**Comments from Reviewer #2**

My only concern, associated to a minor revision, is that this study makes use of other measurements acquired during the deployment period, and more specifically on measurements from the HALO flights. I think that it would be very useful to have a table that lists the instruments (and the geophysical variables from these instruments) onboard the HALO. Such list, provided ahead of the examples (section 4) would help to see what kind of synergies, and thus what kind of analyses, can be done thanks to them.

We understand this point, however we have the feeling that the comprehensive list of instruments onboard of HALO (see Konow et al., 2021) and other possible platforms, like the research vessels equipped with meteorological radar systems, would go beyond the scope of a data paper describing Poldirad. We have added a paragraph at the beginning of section 4 pointing to additional references:

Synergies with other observations are expected in future evaluation of the EURECC4A campaign. An example of synergy can be found in Stevens et al. 2021 (their Fig. 11). Comparisons with further radar observations (e.g. Acquistapace et al., 2021, Konow et al., 2021) are anticipated. All observational system which were available during the campaign are described in Stevens et al. (2021).

Additional references:

Acquistapace, C., Coulter, R., Crewell, S., Garcia-Benadi, A., Gierens, R. T., Labbri, G., Myagkov, A., Risse, N., and Schween, J. H.: EUREC[4]A's Maria S. Merian ship-based cloud and micro rain radar observations of clouds and precipitation, Earth Syst. Sci. Data Discuss. [preprint], https://doi.org/10.5194/essd-2021-265, in review, 2021.

Stephan, C. C., Schnitt, S., Schulz, H., Bellenger, H., de Szoeke, S. P., Acquistapace, C., Baier, K., Dauhut, T., Laxenaire, R., Morfa-Avalos, Y., Person, R., Quiñones Meléndez, E., Bagheri, G., Böck, T., Daley, A., Güttler, J., Helfer, K. C., Los, S. A., Neuberger, A., Röttenbacher, J., Raeke, A., Ringel, M., Ritschel, M., Sadoulet, P., Schirmacher, I., Stolla, M. K., Wright, E., Charpentier, B., Doerenbecher, A., Wilson, R., Jansen, F., Kinne, S., Reverdin, G., Speich, S., Bony, S., and Stevens, B.: Ship- and island-based atmospheric soundings from the 2020 EUREC[4]A field campaign, Earth Syst. Sci. Data, 13, 491–514, https://doi.org/10.5194/essd-13-491-2021, 2021.